# Remote and precise control over morphology and motion of organic crystals by using magnetic field

Xuesong Yang[1], Linfeng Lan[1], Liang Li [2,3], Xiaokong Liu[1], Panče Naumov [2,4✉] & Hongyu Zhang[1✉]

Elastic organic crystals are the materials foundation of future lightweight flexible electronic, optical and sensing devices, yet precise control over their deformation has not been accomplished. Here, we report a general non-destructive approach to remote bending of organic crystals. Flexible organic crystals are coupled to magnetic nanoparticles to prepare hybrid actuating elements whose shape can be arbitrarily and precisely controlled simply by using magnetic field. The crystals are mechanically and chemically robust, and can be flexed precisely to a predetermined curvature with complete retention of their macroscopic integrity at least several thousand times in contactless mode, in air or in a liquid medium. These crystals are used as optical waveguides whose light output can be precisely and remotely controlled by using a permanent magnet. This approach expands the range of applications of flexible organic crystals beyond the known limitations with other methods for control of their shape, and opens prospects for their direct implementation in flexible devices such as sensors, emitters, and other (opto)electronics.

[1] State Key Laboratory of Supramolecular Structure and Materials, College of Chemistry, Jilin University, Changchun 130012, P. R. China. [2] Smart Materials Lab, New York University Abu Dhabi, PO Box 129188 Abu Dhabi, UAE. [3] Department of Sciences and Engineering, Sorbonne University Abu Dhabi, PO Box 38044 Abu Dhabi, UAE. [4] Molecular Design Institute, Department of Chemistry, New York University, 100 Washington Square East, New York, NY 10003, United States. ✉email: pance.naumov@nyu.edu; hongyuzhang@jlu.edu.cn

Flexibility is one of the main assets of any material, and it is one of the key factors that determines its mechanical compliance. It has been demonstrated recently in some common crystalline materials that have been normally considered stiff and hard, such as water ice[1], natural minerals[2], sponge spicules[3], and even nanosize pillars of one of the hardest known materials, diamond[4–6]. Although being micro/nanocrystalline, all of these materials, however, have limited applications in technology and practice, and at the current stage the interest in their flexibility is primarily driven by a scientific curiosity. Much more relevant to applications, especially in view of flexible (opto)electronics, are organic crystalline materials, and specifically flexible organic crystals that—within the realm of the new research direction of crystal adaptronics[7–12]—comprise a new addition to the global materials' property space[13–15]. It has been recently established that many organic crystals, when having sufficiently high aspect ratios, exhibit flexibility and become elastically or plastically deformable. Although the mechanistic aspects of these observations have been and remain a subject of an ongoing debate[16], their application prospects are undisputed. Indeed, the long-range ordered lattices of these materials are thought to carry a tremendous potential for electronics and actuating applications, for example, in flexible wearable devices. At a more fundamental level, the flexibility of organic crystals is thought of as the key to the development of engineering systems whose shape can evolve over time, while they remain homogeneous in composition, ordered in structure, and light in weight. External control over such shape-shifting of organic crystals has been already demonstrated by using light[17–19], heat[20,21] and mechanical force[22,23], and most recently, by a spatiotemporal gradient in humidity[24]. However, it is to be noted that some of the underlying processes induced by these effectors are accompanied by internal structural change (light and heat), and may also result in mechanical damage and short lifetime (force), or slow response (humidity). An even more challenging although oftentimes overlooked aspect of these materials, however, appear to be the difficulties with control over the degree of their deformation. Even that elaborate mathematical models have been developed to describe simple crystal deformations such as bending or twisting, these processes can be easily initiated, but cannot be terminated at will[9,25]. As a result of the fact that the mechanical deformation occurs as a release of internal stress and at a much slower time-scale than the action of the stimulus, at present, even simple deformations such as crystal bending remain highly uncontrollable in view of the degree of bending and the final shape of the crystal.

Along this line of pursuit, we have recently embarked on a research to explore alternative means that would enable not only remote, but also precise control over the crystal shape. Magnetic materials have been widely used in spatial control and actuation, and are at the core of technologies that range from demonstrations of simple delivery of cargo in microrobotics to smart systems for biological separations[26–30]. As demonstrated with applications in medical diagnostics and therapy, biotechnology and bionic research, the operational distance, dexterity, precision, speed and robustness with remote magnetic manipulation of small objects are superior to other means of object manipulation[31–33]. The permanent magnetic field is a powerful means that enables controlled kinematics, precise dislocation of objects, and rapid energy transfer[34–36]. Building on the success of magnetic field in the control of shape of polymers[37–40], we have recently developed and describe here an approach that uses magnetic field to effectively control the shape of flexible organic crystals without the need of direct physical contact or exposure of the crystals to light, heat or humidity. Briefly, flexible organic crystals were first coated by two polymers, and then combined with magnetic nanoparticles. The resulting hybrid materials respond to magnetic field remotely and rapidly, and they can be morphed with very high precision both in 2D and 3D spaces. Coating of organic crystals was used recently as a method to prevent their dissolution in organic solvents[41]. Our approach to attain precise spatial control over the deformation of crystals is simple, universally applicable, and circumvents the known drawbacks when using light, heat or force to induce motion or deformation. As a proof-of-concept, we describe application of the hybrid crystals as microoptical waveguides where the signal output can be precisely controlled. The results suggest that this technology holds potentials for flexible optoelectronics and bionics in both aerial and underwater optical signal transduction.

## Results and discussion

**Preparation of the magnetic hybrid flexible crystals.** Six organic small molecule compounds, shown as **1–6** in Fig. 1a, were used to prepare the magnetic hybrid materials. The crystal structures of compounds **2**, **4**, **5** and **6** were reported previously[20,24,42,43]. While compound **1** is newly synthesized, compound **3** was reported without single crystal diffraction data[44]. The structures of both compounds were determined, deposited, and are available from the CCDC (codes: 2116768 and 2117194; Supplementary Figs. 1–7, Supplementary Table 1). Acicular crystals of all compounds were grown by solvent evaporation (Supplementary Fig. 8). Their lengths were at the order of centimeters, and their widths and thickness were at the order of tens to hundreds of microns (Supplementary Table 2). As shown in Fig. 1b, when subject to external force the crystals were elastic and could be bent without breaking. The mechanical properties of the crystals of **1** and **3** were characterized, and the results are available from Supplementary Fig. 9a and 9c. From the studied materials, only crystals of **4** could be bent on two of the crystals' faces (Supplementary Fig. 10)[43]. This elasticity can be tentatively attributed to the extensive intermolecular interactions in their structures, such as hydrogen bonding and π—π interactions. The method for preparation of the magnetic hybrid crystals is shown schematically in Fig. 1c, d. The hybrid crystals were expected to respond when approached by a permanent magnet (Fig. 1e). To that end, the nascent organic crystals were first coated with multiple alternating polymer layers by alternative treatment with aqueous solutions of a positively charged polymer, poly(diallyldimethylammonium chloride) (PDDA) and a negatively charged polymer, poly(sodium 4-styrenesulfonate) (PSS). Typically, eight layers of each polymer were applied. The polymer-coated crystals were then affixed at both ends to a capillary glass tube (Supplementary Fig. 11), and a suspension of iron(III) oxide magnetic nanoparticles (MNPs) was aspirated with a syringe. One part of the polymer-coated crystals was spin-coated using the droplets at the tip of the syringe, whereby partially MNP-coated crystals of MNP//**1–6** were obtained which had a small strip with MNP closer to one end (Fig. 1f). Figure 1g shows a scanning electron micrograph of a cross-section of the crystal, and Fig. 1h represents the cross-section of the magnetic hybrid flexible crystals MNP//**1–6** schematically. To test whether there is significant alteration of the mechanical properties of the crystals after the coating with polymer and application of MNPs, the mechanical properties of crystals of **1**, **3**, MNP//**1**, and MNP//**3** were determined and compared (Supplementary Fig. 9). The results confirmed that the mechanical properties of the hybrid organic crystals were essentially unchanged relative to the original crystals.

In order to attain control over the thickness and length of the magnetic layer, aqueous suspensions with varying concentrations of iron(III) oxide were tested on selected crystals of **1**, **3** and **5**. It was found that the uniformity of the MNPs on the crystal surface improved greatly at higher concentration of the suspension. Crystals of **1**, **4** and **5** were selected to study the length of the

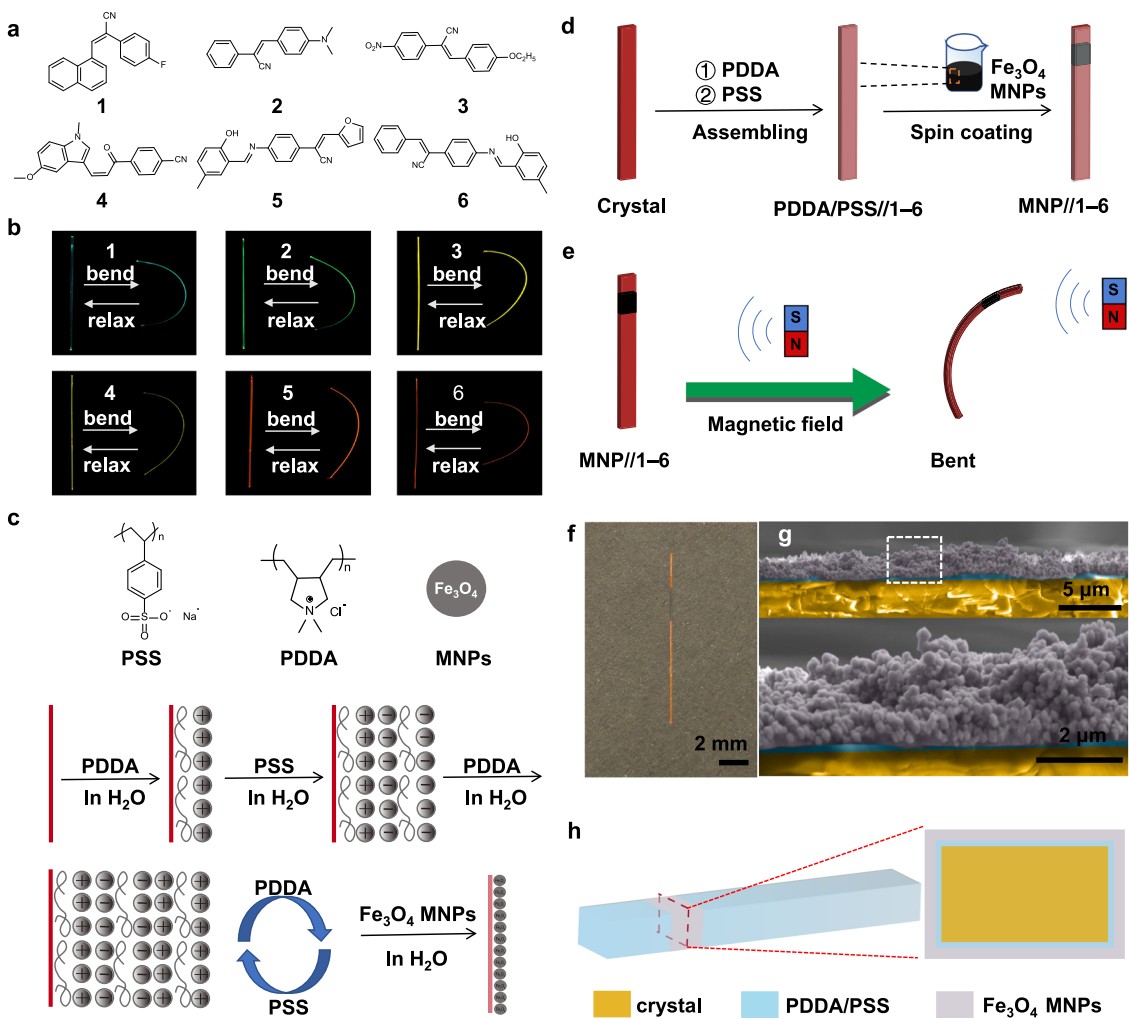

**Fig. 1 Concept and preparation of magnetic hybrid flexible crystals. a** Chemical structures of elastic crystals **1–6**. **b** Reversible bending and unbending of acicular crystals **1–6**. The process was observed by using the emission of these crystals under 365 nm UV light for better contrast with the dark background. **c** A procedure for charge-driven assembly of magnetic nanoparticles (MNPs) onto flexible organic crystals (Red vertical lines represent flexible organic crystals). **d** Preparation of the hybrid crystals MNP//**1–6**. **e** Schematic diagram showing the mechanism of bending of MNP//**1–6** in a magnetic field. **f** Photograph of MNP//**5** in daylight. **g** False-colored scanning electron micrographs showing the cross-section of the crystal hybrids. The crystal is shown in orange color, the polymer coating is in blue, and the Fe$_3$O$_4$ MNPs are in light grey color. **h** Schematic cross-section of the magnetic hybrid flexible crystals MNP//**1–6**.

MNP strip on the crystal. MNP strips of length between 1 mm and 2 cm were deposited on the surface (Supplementary Fig. 12). Since at concentrations of 2 g mL$^{-1}$ and 3 g mL$^{-1}$ the suspension was particularly dense and homogeneous (Supplementary Figs. 13–15), subsequent experiments employed hybrids obtained by using an aqueous solution of MNPs with concentration of 2 g mL$^{-1}$ and lengths of the MNP strip of about 1 mm. The thickness and homogeneity of the MNP coating on the surface of MNP//**1**, observed by scanning electron microscopy, revealed that the magnetic layer was 2.43 ± 0.45 μm thick, and it was usually approximately 1/40 of the crystal's thickness (Supplementary Fig. 16).

**Accurate control of the bending and deformation of the magnetic hybrid flexible materials**. The presence of the MNP-coated fragment on the crystal surface enables control over their shape simply by using an external magnetic field. This can be easily demonstrated by fixing one end of the MNP//**1–6** to a surface (in our case, a silicon wafer) and by moving a magnet beneath the surface to change the position of the free end of the crystal. Figure 2a illustrates the change of the originally straight

crystal to an "omega"-like shape. In another exemplary illustration, Fig. 2b shows reconfiguration of the crystal by the same method into various numbers, letters, and other familiar shapes. As also shown in Fig. 2b, this method can be used to simultaneously manipulate a group of crystals, and could therefore be applied to assemble multiple crystals and/or to simultaneously reshape them. By using magnets, macroscopic motion of the crystal can also be induced. To that end, two strips of MNPs were deposited on both ends of a crystal. The crystal was placed on a silicon wafer, and its motion over the wafer was induced and sustained simply by changing the position of the magnets below the wafer (Fig. 2c, Supplementary Movie 1). Such a crystal can be considered equivalent to a biomimetic actuator, and was able to move over a distance of up to 7 cm in 18 seconds. These visually appealing examples of magnetically induced morphing and motility induced in the hybrid materials illustrate the potential of using magnetic field with any flexible organic crystals for remote and arbitrary change in their shape with a high degree of control over that process.

The manipulation of the crystal shape by using a magnetic field described above comes with a control over the shape of the crystal

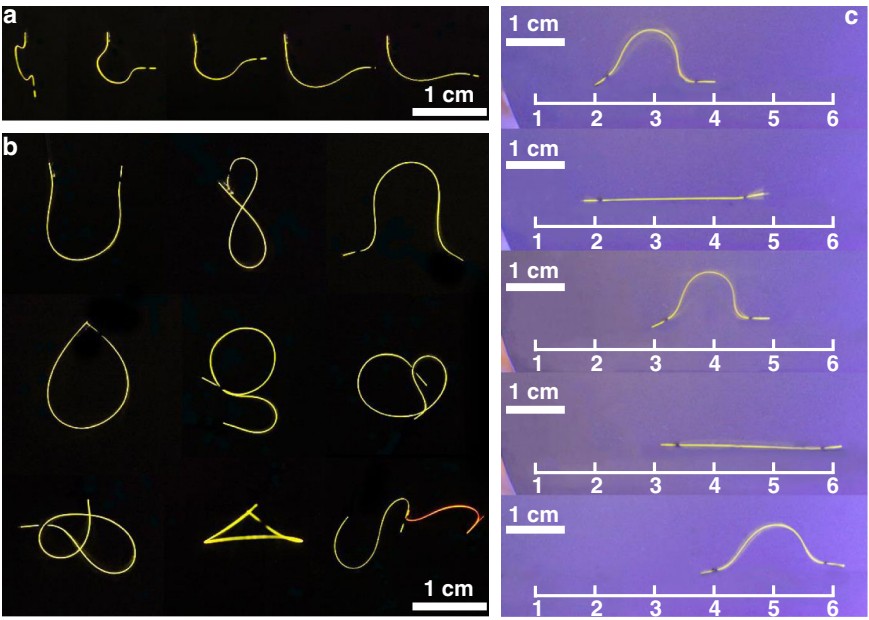

**Fig. 2 Shape-shifting and motility of hybrid magnetic organic crystals. a, b** Photographs of MNP//**4** and MNP//**5** that were reshaped by using magnetic field observed under UV light. Note that the contrast of the color in these images was enhanced for clarity. **c** Example of an inchworm-like motility of MNP//**4** induced by a temporally evolving spatial gradient in the magnetic field. The white scale bar corresponds to a length of 1 cm. Note that the contrast of the color in these images was enhanced for clarity.

in a plane or in space. To demonstrate the change in shape in two dimensions, individual samples of the hybrid crystals were affixed at their one end on the surface of a silicon wafer with glue (Fig. 3a). A total of 9 points were selected at angles that are within the elastic regime of the crystal (=180°; Fig. 3b), and the bending angle of the organic crystal could be precisely controlled by simply moving the magnet under the silicon wafer. As shown with two examples (MNP//**4** and MNP//**6**) in Fig. 3c and Supplementary Fig. 17, the crystals were deflected and their ends were aligned exactly at the set points, which enables precise control over their curvature. This control in shape is not limited to a plane. A crystal of MNP//**4**, which was found to be capable of bending on two different crystal faces, was suspended into a round glass vessel, and a magnet was moved across the wall of the container. The crystal responded promptly by bending towards the magnet and could be pointed at an arbitrary direction (Fig. 3d, Supplementary Movie 2). This control over the crystal motion was reproduced after filling the vessel with water (Fig. 3e, Supplementary Movie 3). These experiments show that the hybrid crystals are capable of fast and arbitrary reshaping with a precise curvature in air as well as in a liquid medium, even that such control over the shape and dynamics of deformation is not possible by using other means. Bending of the crystals in a liquid medium could be important, among others, in underwater sensing applications.

Durability over prolonged use and resistance to fatigue and wear are some of the most essential prerequisites for practical applications of organic crystals as optoelectronic devices or sensors. When using light or heat, the crystals inevitably accumulate defects, resulting in decreased performance over time. The reproducibility of the crystal deformation and fatigue was tested by a device shown schematically in Fig. 3a, b. A crystal of MNP//**4** on a silicon wafer was bent by using a magnet that was shifted back and forth between two points (−90° and 90°) under the wafer. The crystal of MNP//**4** pointed precisely at each point under the control of the magnet even after 3000 cycles (Fig. 3f). Moreover, the crystal showed high durability at moderately higher temperatures (50 °C) (Supplementary Fig. 18). In addition,

when the crystals of MNP//**5** were placed in water for different periods of time, they were still able to bend by attraction by the magnet (Supplementary Fig. 19). This test shows that since the bending is not caused by an intrinsic chemical reaction or a thermal process, it only depends on the flexibility of the crystal, and it could be in principle repeated indefinitely. If the crystal is stable in air, this process can be repeated even after a stationary period over a long time. We confirmed, for example, that the above crystal could still be bent precisely at the set point even after three months.

**Application of the magnetic hybrid flexible crystals as spatially controllable waveguides.** As only one demonstration of the plethora of possible future applications that could be envisaged for these hybrid flexible crystals we studied their performance as optical waveguides. Recently, flexible organic crystals have been recognized for their great potential as optical transducing medium, both in the visible[7,45] as well as in the near-infrared (telecom)[46] range, and there has been a burgeoning research into their implementation in optical microdevices. Reshaping of these waveguides by using UV light[19] or humidity[24] has also been reported. At a microscopic scale, these optical waveguides can be precisely assembled into optical circuits by using the tip of atomic force microscope[13,14,47,48]. Considering the control over the crystal shape discussed above, we aimed to demonstrate the application of the hybrid crystals as multidirectional, ultraprecise, and ultradurable optical waveguides that can be reconfigured both in two and three dimensions by using magnetic field (Fig. 4a, b).

As shown in Fig. 4c, a crystal of MNP//**4** was excited with 355 nm UV light, and its end which acts as optical output was moved with the magnet. To verify whether the coating of $Fe_3O_4$ MNPs on the crystal surface affects the waveguiding properties, the optical loss coefficients (OLCs) of the original crystal **4** and MNP//**4** were both tested and compared as active optical waveguides. As expected, the emission intensity gradually decreased with distance from the irradiation position (Fig. 4d–f)

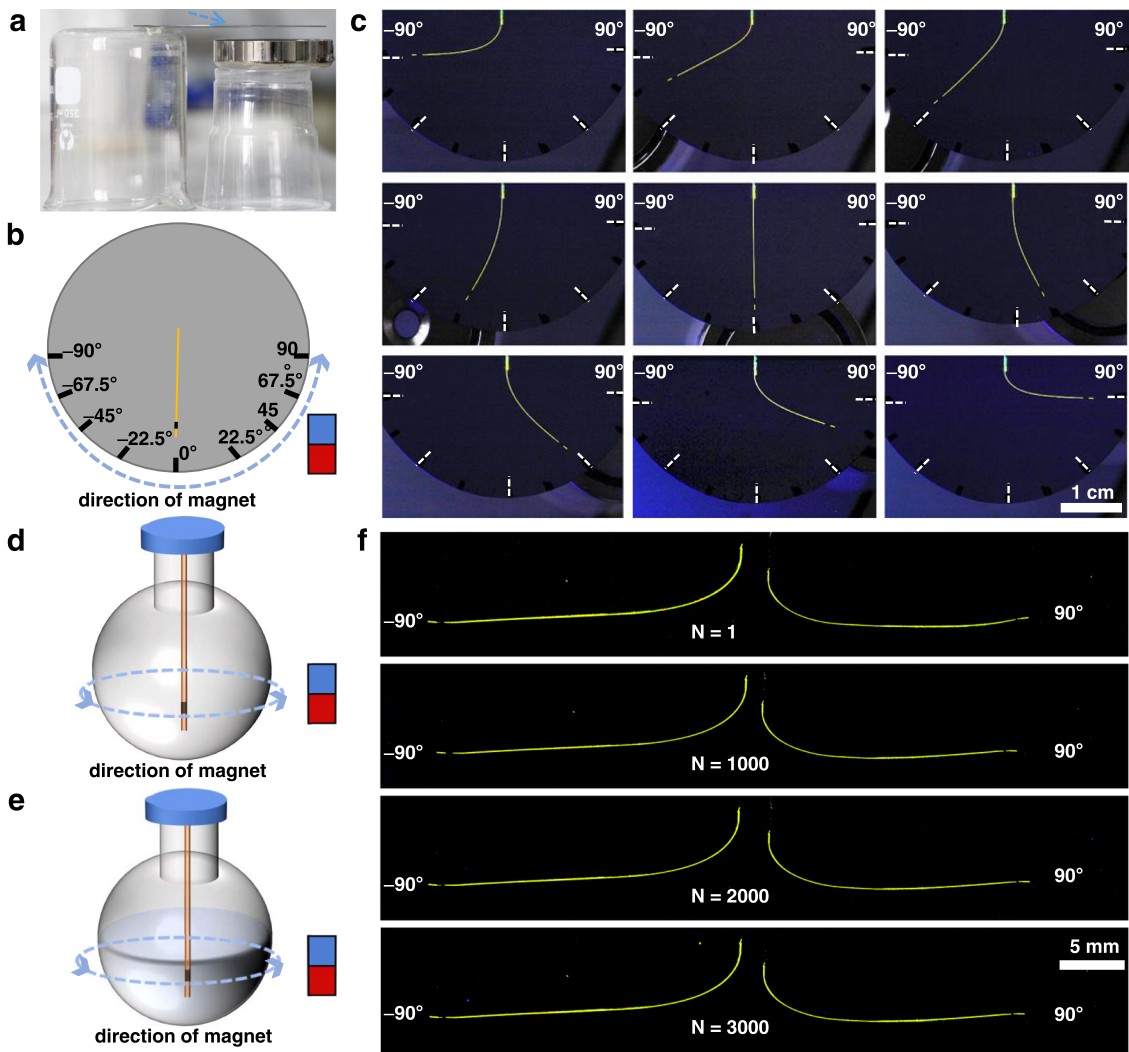

**Fig. 3 Spatial control over the shape of the hybrid crystals by using magnetic field. a** Optical photograph of an MNP//**1–6** crystal bent at an arbitrary angle within a plane (side view). The blue arrow indicates the position of the crystal. **b** Schematic diagrams of MNP//**1–6** bending at an arbitrary angle in a plane. **c** Photographs of a crystal of MNP//**4** bending precisely towards 9 points that were marked on a silicon wafer. Note that the contrast of the color in these images was enhanced for clarity. The blue arrow represents the direction of movement of the magnet. **d**, **e** Schematic diagrams of MNP//**1–6** bending at an arbitrary angle in any direction in space (**d**), and in any direction in space inside a liquid (**e**). The blue arrow represents the direction of movement of the magnet. **f** Photographs of the MNP//**4** crystal at different times in a durability test. During the test the crystal was bent between –90° and 90° by moving the magnet under the silicon wafer, followed by returning it to –90°. Note that the contrast of the color in these images was enhanced for clarity.

due to the optical loss. The distance-dependent emission spectra were obtained by using the same light (355 nm laser) to excite the crystal at different positions and the emission spectra were recorded at the end of the crystal (Fig. 4g–i). By fitting this data according to a procedure reported earlier[49] the OLCs at the emission peak were calculated to be 0.256 dB mm$^{-1}$ for **4** in the straight state, 0.263 dB mm$^{-1}$ for MNP//**4** in the straight state, and 0.267 dB mm$^{-1}$ for MNP//**4** in the bent state (Supplementary Fig. 20). Since the OLCs of MNP//**4** in the straight state were only slightly higher than those of **4** in the straight state, we conclude that the optical transmission of **4** was practically unaffected by the MNPs. To check the generality of these conclusions, the optical transmission of the native and hybrid crystals **2** and **3** in their straight state as active optical waveguides were also recorded (Supplementary Fig. 21). The OLCs of the straight crystals of **2** and **3** are 0.098 dB mm$^{-1}$ and 0.130 dB mm$^{-1}$, respectively, and the respective values for MNP//**2** and MNP//**3** are 0.106 dB mm$^{-1}$ and 0.139 dB mm$^{-1}$ (Supplementary Fig. 22). In addition, MNP//**4** can be used as a passive optical waveguide for optical signal

transmission (Supplementary Fig. 23). These results confirm that the hybrid crystals are capable of multidirectional transduction of light both in two and three dimensions, and that the light output can be controlled by using magnetic field.

**Relating magnetic field strength and bending of magnetic hybrid flexible crystals.** We were also interested in whether there is a relationship between the magnetic field strength and the degree of bending of the crystal. A setup shown in Fig. 5a (Supplementary Fig. 24) was used to examine this hypothesis. A hybrid crystal of MNP//**4** was affixed at one end with glue, and it was slowly approached with a permanent magnet from one side perpendicular to its longest axis, while recording the magnetic field strength ($B$) with a probe connected to a Gaussmeter from the other side. In two experiments, the bending of the crystal was found to increase with increasing magnetic force (Fig. 5b–e). In order to test reproducibility, the position of the detection probe on the Gaussmeter was changed, and the measurements were

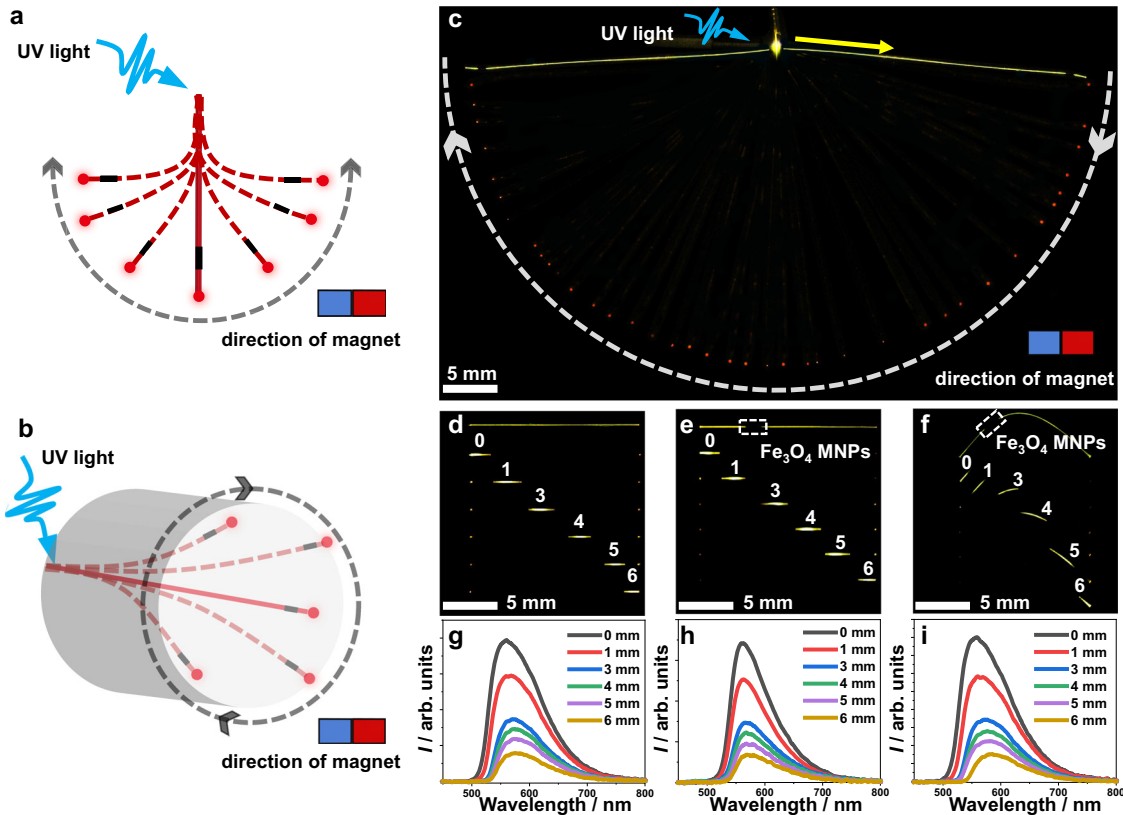

**Fig. 4 Hybrid organic crystals as magnetically controllable organic crystal optical waveguides. a, b** Schematic of the experiments that demonstrate control over the shape of a crystal as optical waveguide by using magnetic field in two dimensions (**a**) and in three dimensions (**b**). The grey arrow represents the direction of movement of the magnet. **c** A crystal of MNP//**4** acting as magnetically controllable optical waveguide. One end of the crystal was affixed to the base while the other was free and its position was controlled by moving the magnet. The crystal was excited by a 355 nm laser light (blue arrow). The yellow arrow indicates the direction of light propagation, and the white arrow shows the motion of the magnet and the tip of the crystal. The red dot is the light output. Note that the contrast in this figure has been enhanced for clarity of inspection of the light output. **d–f** Images of the crystal of **4** used as a waveguide in uncoated the straight state, **4** (**d**), coated straight state, MNP//**4** (**e**) and coated bent state (**f**). **g–i** Fluorescence spectra collected at one tip of the crystal with different distances between the tip and the excitation site of the 355 nm light. The spectra in panels g, h and i correspond to the crystals shown in panels d, e and f, respectively.

repeated by using MNP//**6** (Supplementary Fig. 25). The results confirmed the relationship between the degree of bending of MNP//**6** and the strength of the magnetic field. This observation provides further opportunities into modulation of the crystal deformation by using variable magnetic field.

In summary, we report a non-destructive method for non-contact bending or motion of surface-modified flexible organic crystals. The use of magnetic field described here surpasses—both in the degree of spatiotemporal control, kinetics of bending and robustness over prolonged operation—all of the previously reported approaches using other stimuli to exert control over the crystal shape, such as light, heat or humidity. The Fe$_3$O$_4$ MNPs used here are chemically stable, and the method for their attachment to the crystal provides a robust device that does not disintegrate or deteriorate over time. We confirmed that the MNPs adhere well and do not separate from the crystals during the deformation. With a flexible crystal of good quality and such that is sufficiently elastic so that it does not fracture over many cycles of bending within its elastic regime, the hybrid crystals can be bent thousands of times with complete retention of their integrity and with sustained precision in their curvature and shape. This approach significantly expands the application prospects of flexible crystals beyond the existing limitations with other methods, and favors their application in flexible devices such as sensors and wearable devices. Moreover, the strategy proposed to assemble polymers capable of different responses on

the outer layer of flexible organic crystals can be a general approach to preparation of active sensing elements for detection of gases or biosensors, among other applications.

## Methods

**Materials.** All solvents and starting materials for the syntheses were purchased from commercial sources and were used as received. Poly(diallyldimethylammonium chloride) (PDDA, MW ca. 200000–350000), poly(sodium 4-styrenesulfonate) (PSS, MW ca. 70000), Fe$_3$O$_4$ MNPs (approximate size: 50 nm) and NaCl were purchased from Energy Chemicals Co. The synthesis of compounds **1−6** is provided as Supplementary Figs. 1–6, where compounds **2**, **3**, **4**, **5** and **6** were synthesized according to literature procedures (Supplementary Figs. 26–37)[20,24,42–44]. Photographic images of crystals of **1−6** and details on their preparation are provided as Supplementary Fig. 8.

**Fabrication of magnetic hybrid flexible crystals MNP//1–6.** The organic crystals were immersed in an aqueous solution of 1 mg mL$^{-1}$ poly(-diallyldimethylammonium chloride) (PDDA) for 20 min, followed by 1 min rinse with distilled water. Then the crystals, having positively charged surfaces, were immersed in an aqueous solution of 1 mg mL$^{-1}$ poly(sodium 4-styrenesulfonate) (PSS) for 20 min, and then rinsed with distilled water for 1 min. By repeating the above steps, the crystals were coated with multiple alternating polymer layers. The ends of such polymer-coated crystals were fixed to a capillary glass tube, and a suspension of iron(III) oxide nanoparticles was aspirated with a syringe. Part of the polymer-coated crystals was then spin-coated using the droplet at the tip of the syringe to obtain the hybrid crystals MNP//**1–6**.

**X-ray crystallographic analysis.** Single crystal diffraction data for compounds **1** and **3** were collected on a Rigaku Synergy diffractometer with monochromated

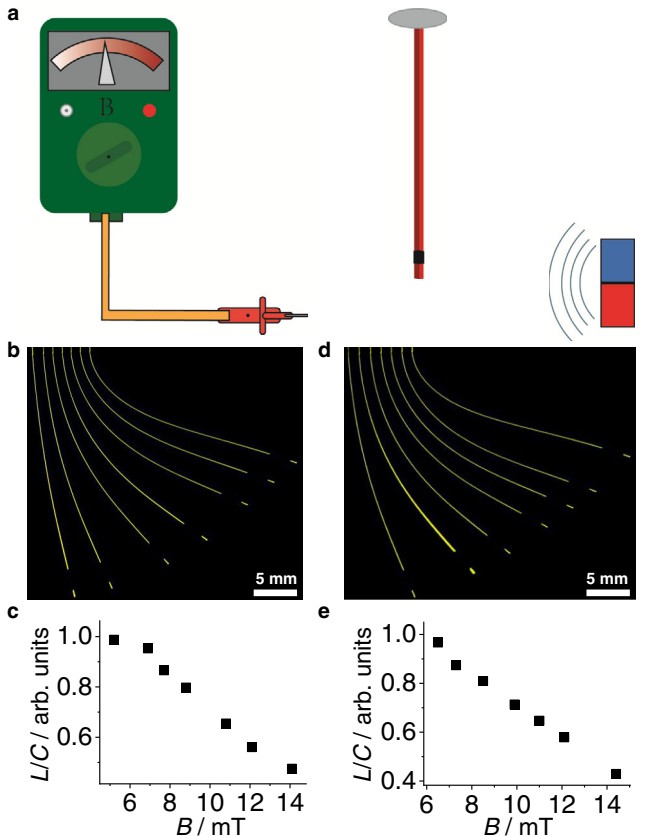

**Fig. 5 Experimental setup and results of effect of the magnetic field on crystal bending. a** Schematic of the experimental setup. The magnet approaches a vertically standing crystal from the right side, and the deflection is recorded on the opposite side. **b**, **d** Images showing the bending of crystals of MNP//**4** in the test (**b**) and (**d**) while they were being approached with the magnet. **c**, **e** The ratio of the chord length ($L$) and arc length ($C$) of the crystal plotted against the magnetic field strength ($B$).

MoK$_\alpha$ radiation ($\lambda = 0.71073$ Å). The CrysAlis Pro software was used for data collection, integration, scaling and absorption correction. The data was corrected for absorption effects by using SADABS[50]. The structure determination was performed by using the OLEX2 interface[51], and the structure refinement was performed by using the full matrix least-squares method, based on $F^2$ against all reflections with SHELXL-2014/7[52]. The non-hydrogen atoms were refined anisotropically. The positions of the hydrogen atoms were calculated and refined isotropically. The images of the structures were generated by using Mercury 4.3.1[53]. Additional details and crystallographic data are provided in Supplementary Table 1.

## Data availability

The crystal structures reported in this article have been deposited and are available free of charge from the Cambridge Crystallographic Data Centre (CCDC), with numbers 2116768 for **1** and 2117194 for **3**. All data are available from the corresponding authors upon request.

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

## Acknowledgements

This work was supported by the National Natural Science Foundation of China (51773077 H.Z. and 52173164 H.Z.), the Jilin Province International Collaboration Base of Science and Technology (YDZJ202102CXJD004 H.Z.) and a fund from New York University Abu Dhabi (P.N.).

## Author contributions

X.Y., P.N., and H.Z. conceived the study. X.Y. prepared and performed the experiments on the hybrid crystals. X.Y. performed experiments related to shape change, multi-directional bending, durability tests, and the relationship between the degree of bending and magnetic field. X.Y. performed experiments related to optical waveguiding properties under guidance of LF.L. X.L. provided experimental setups. X.Y. revised the manuscript under guidance of L.L. The manuscript was written with contributions from X.Y., L.L., P.N., and H.Z. All authors have given their approval for the final version of the article.

## Competing interests

The authors declare no competing interests.
