## [Peer Review File · Nature Communications]

REVIEWER COMMENTS

Reviewer #1 (Remarks to the Author):

This manuscript reports a non-destructive method for non-contact bending or motion of surface-modified flexible organic crystals. The author puts the organic crystals into the aqueous solutions of the positively charged polymer PDDA and the negatively charged polymer PSS, respectively, and coats them with multiple alternating polymer layers. Then the author obtained partially MNP-coated crystals by spin-coating one section of the polymer-coated crystals with iron(III) oxide magnetic nanoparticles (MNPs). These MNP-coated crystals have fascinating properties. They can easily control their shape by using an external magnetic field, which can be used for flexible optical waveguides and magnetic sensing. This work was well done and novel enough to be published in Nature Communications after paying attention to the following minor points:

1. As for flexible crystals, it is necessary to characterize their mechanical properties, such as its Young's modulus. And the mechanical properties of the flexible crystals before and after polymer and MNP treatment should also be compared.
2. Figure 1h is the same as Figures S7b and 7C. Please explain why it appears twice, or delete the figure in the supporting information.
3. In page 8 lines 172, the author said that "The crystal did not show any fatigue or defects, and pointed precisely at each point even after 3000 cycles (Figure 3f)." In fact, if the crystal is fatigued or defective, it can still point precisely at each point due to the presence of the flexible polymer layer coated on the outside. Therefore, can the author provide further evidence to show that 3000 cycles does not affect the crystal?

Reviewer #2 (Remarks to the Author):

In the submitted article, the authors elegantly present an original attempt to integrate organic crystals into flexible actuators that can be remotely controlled using magnetic fields. Over the last decades, the field of the organic crystals popped-up with materials exhibiting interesting physical properties that paved the way towards potential brand-new actuators without clearly getting to the end of the fabrication and qualification of such devices. In the present paper, the authors successfully and elegantly bridge this gap and confirm the possibility to implement such crystals in conventional process flows that approach the standards of the traditional microfabrication techniques realm.

As a preliminary observation I would suggest to the authors not to use the “soft robot” terminology; for an artefact to be qualified as a robot, it has to have the capacity to sense its environment, process the received information and act as a consequence of it. Instead of “robot”, I would use the “actuator” term, especially in the context of the paper.

I would split my report in three parts: experimental protocols, quantitative analysis and suggestions to improve the added value of the paper.

Experimental protocols

All the chemistry and instrumentation protocols are thoroughly described either in the main article and in the supplementary information files. The fabrication of the actuators is clearly stated. It would be useful to provide the details of the magnetic particles spin coating protocol.

Six different organic compounds are used all over the experiments. Nevertheless, only some of them seem to be selected to perform some of the experiments (e.g. MNP//4 is used for the inchworm-like motility demonstration, MNP//4 again, as well as MNP//6, are used to somehow quantify the linear relationship between the magnetic force and the deformation of the hybrid structures). Could the authors comment on these selections? Is the MNP//4 structure more chemically responsive than the others?

Quantitative analysis

An attempt to quantify the dependence of the hybrid structures deformation with respect to the magnetic field is given in the last part of the article and it clearly shows a linear dependence. What is less clear is the correlation, for instance, in between the positions shown in figure 5.b (eight of them) and the points plotted in figure 5.c (seven of them). Meanwhile, the extreme deformations would suggest a non-linear behavior of the tested structure while the plots still exhibit linear shapes. Could the authors be a bit more precise in their explanations regarding this part?

What would be interesting is to have an estimation of the effective Young modulus of the hybrid crystals by measuring the deformation and estimating the moment of inertia of the structures. This value is expected to be lower than the Young modulus of the sole crystals. I do not request these supplementary tests and data here as this is not the main scope of the paper. I am sure, given the high applicative potential of the presented structures, that a thorough qualification of actuation performances paper could come in the next future.

Suggestions to improve the added value of the paper

Answering the questions asked above in the main body of the article will for sure improve its readability. My last comment will concern the optical waveguide part which is also well discussed and opportunely presented as one possible application. I would invite the authors to think bigger and eventually comment this in their concluding part: what if one the polymers (say, the external layer) is used as gas or biological sensor? The authors have already a device which has integrated actuation (magnetic) and sensing (optical) schemes which, by means of a well chose polymer, could be already used as biological or chemical ready for real world applications. This is, I believe, a huge added value of an already complete, well written, thoroughly documented and original paper which deserve publication in Nature Communications review.

Reviewer #3 (Remarks to the Author):

Dear Editor,

In this communication by "Anonymous" et al., the author/ authors present a novel and innovative way of bending slender and elastic molecular crystals using magnetic fields. As mentioned in the abstract and introduction, some molecular crystals exhibit elastic properties that allow them to be deformed via force, heat, or photons in the case of photomechanically responsive crystals. In this reviewer's opinion, deforming crystals by light is still a superior method for shaping crystals since light can be focused, change wavelength, polarity, and intensity all in a flip of a switch, and can act on select regions of a crystal from unlimited distances without losing power, if properly focused. However, photomechanically responsive molecular crystals are scarce with larger photomechanical crystals that tend to fracture due to the buildup of interfacial stress between the photoproduct layer and reactant. In some cases, light cannot penetrate deep enough in the crystal due to the formation of an absorbing photoproduct layer. This paper presents a "general" method for coating cm-long acicular molecular crystals with alternating layers of poly(diallyldimethylammonium chloride) (PDDA) and a negatively charged polymer, poly(sodium 4-styrenesulfonate) (PSS). This composite layer acts as a platform onto which iron(III) oxide magnetic nanoparticles (MNPs) layer can be deposited. The method of coating these crystals with polymers followed by nanoparticles is subtle yet innovative. By coating, only one section or alternating sections of the molecular crystals studied they have created a truly hybrid new material that mechanically responds to a magnetic field. The mechanical response is large enough to be visualized with the added property of being fatigue-resistant over thousands of cycles at moderate temperatures. The mechanical response was assessed qualitatively and quantitatively in air and water by varying the magnetic field intensity and measuring the deflection angle.

The abstract is concise and reflects the claims mentioned in the text and conclusion. The method used to measure the deflection angles was clearly stated in the experimental section. I highly recommend this for publication as communication in Nature Communications, however, there are some minor inconsistencies and suggestions that need to be addressed to improve the quality of this paper.

Please consider incorporating the following reference:

a) "Chiba H, Morimoto M, Irie M. Stepwise Assembly of Ultrathin Poly (vinyl alcohol) Films on Photoresponsive Diarylethene Crystals. Chemistry Letters. 2021 Jan 5;50(1):84-6." In this paper, Irie uses PVA to coat photoresponsive molecular crystals. However, his objective from coating these crystals was to prevent them from dissolving in organic solvents and not to detect water vapor. However, it is worth mentioning that people have thought of coating crystals with a polymer PVA before.

Other issues that need to be addressed:

1. What type of waveguides are these crystals? Are they the active or passive kind? Will the different types of waveguides (active or passive) respond differently in the presence of a polymer and MNP layer on the surface?
2. Decorating select areas of acicular molecular crystals with MNP is applicable only when the crystals are long enough to handle with dexterous fingers. Can this process be automated or applied on much smaller crystals using specialized instruments such as AFM?
3. Are all the crystals studied isotropically elastic? It is worth highlighting that some crystals can be elastic along one crystal direct and not the other, therefore this method may or may not work on some crystals. This may be an added feature where if a crystal was bent in an undesirable direction it will snap.
4. Please elaborate on how you obtained these acicular crystals. Solvents used concentration, temperature, etc.
5. The fatigue resistance of the crystals was measured at ambient temperatures, was there any attempt to evaluate the resilience of the crystal and MNP layer at a slightly higher temperature?
6. How come the polymer layer and MNPs did not fall off the crystal surface after actuation in water, knowing that the polymer layers are water-soluble? How long can the MNP-decorated crystal remain in the water?

Minor inconsistencies:

1. The geometrical isomer configurations of compounds 4, 5, and 6 depicted in Supplementary Scheme 3 are different from the ones in the main text. Please confirm the proper geometrical isomer from the crystal structure or H NMR.

2. Supplementary Figure 2 was not helpful. This reviewer could not see any difference between the vials.
3. The ^1H NMR spectrum of compound 3 reveals the minor presence of a geometrical isomer. However, elemental analysis of compound 3 shows that it is very pure > 99.3%. Mention in the text that the minor presence of geometrical isomers in compound 3 will not affect the elemental analysis results since both isomers have the same molecular formula.
4. Figure 3a is not clear. Please point out with arrows where the crystal is located.

Conclusion:

This paper is significant for both the crystal and polymer community since the authors were able to use both materials to produce a composite system that is greater than the sum of its parts. This method can be extended to coat photomechanical crystals to yield higher-order adaptronic crystals with more functions. I recommend publication after some minor revisions.

Response to reviewers' comments on the manuscript "Remote and Precise Control over Morphology and Motion of Organic Crystals by using Magnetic Field"

In the PDF version of this response, the review comments are written in *black italic font*, our responses to the comments are written in *blue regular font*, and the changes made to the main text are written in *red regular font*.

Response to the comments from Reviewer #1:

Overall Comment: *This manuscript reports a non-destructive method for non-contact bending or motion of surface-modified flexible organic crystals. The author puts the organic crystals into the aqueous solutions of the positively charged polymer PDDA and the negatively charged polymer PSS, respectively, and coats them with multiple alternating polymer layers. Then the author obtained partially MNP-coated crystals by spin-coating one section of the polymer-coated crystals with iron(III) oxide magnetic nanoparticles (MNPs). These MNP-coated crystals have fascinating properties. They can easily control their shape by using an external magnetic field, which can be used for flexible optical waveguides and magnetic sensing. This work was well done and novel enough to be published in Nature Communications after paying attention to the following minor points:*

Response: We thank the Reviewer for their careful and thorough reading, as well as for assessment of the results presented in our manuscript.

Comment: 1. *As for flexible crystals, it is necessary to characterize their mechanical properties, such as its Young's modulus. And the mechanical properties of the flexible crystals before and after polymer and MNP treatment should also be compared.*

Response: We thank the Reviewers for their suggestion. To respond to this comment, we have characterized the mechanical properties crystals of flexible crystals, and we have also compared the mechanical properties of the original flexible crystals with those of the flexible crystals treated with polymer and magnetic nanoparticles (MNPs), and the results have been to the revised article.

The following text has been added on page 3 line 91 of the revised manuscript:

Added text: "The mechanical properties of the crystals of 1 and 3 were characterized, and the results are available from Supplementary Figure 3a,c. Form the studied materials, only crystals 4 could be bent on two of the crystals' faces (Supplementary Figure 4).⁴³"

New Supplementary Figure 3. Stress-strain profiles of the native and coated crystals. (a) Crystal 1. (b) MNP//1. (c) Crystal 3. (d) MNP//3.

The following text has been added on page 4 line 106 of the revised manuscript:

Added text: “To test whether there is significant alteration of the mechanical properties of the crystals after the coating with polymer and application of MNPs, the mechanical properties of crystals of 1, 3, MNP//1, and MNP//3 were determined and compared (Supplementary Figure 3). The results confirmed that the mechanical properties of the hybrid organic crystals were essentially unchanged relative to the original crystals.”

Comment: 2. *Figure 1h is the same as Figures S7b and 7C. Please explain why it appears twice, or delete the figure in the supporting information.*

Response: We thank the reviewer for bringing this to our attention. This was a result of an unintentional mistake, and in the revised version images 7a and 7b in the Supporting Information are deleted.

New Supplementary Figure 10. Micrographs of the crystal of MNP//1 taken under scanning electron microscope. (a) Cross-section of the crystal, (b–c) Surface of the crystal.

Comment: 3. *In page 8 lines 172, the author said that "The crystal did not show any fatigue or defects, and pointed precisely at each point even after 3000 cycles (Figure 3f)." In fact, if the crystal is fatigued or defective, it can still precisely at each point due to the presence of the flexible polymer layer coated on the outside. Therefore, can the author provide further evidence to show that 3000 cycles does not affect the crystal?*

Response: We thank the Reviewer for the careful reading and for this astute comment. The purpose of these durability tests was to investigate whether or not the crystals can retain their ability to point in any direction (in the plane) with good reproducible and high accuracy after multiple bending cycles induced with the action of a magnet. The statement in the text that "no defects or fatigue were exhibited" was intended to convey this observation. We do concur with the Reviewer that our original formulation might leave some room for interpretation on whether the crystal exhibits defects or fatigue after multiple bending. In order to verify this, two additional experiments were designed and performed to test whether defects or fatigue in the crystals were caused after multiple bending: (1) single crystal X-ray diffraction analysis was performed before and after bending (provides insight into the microscopic changes), and (2) photographs of the crystal were taken before and after bending (provides information on the macroscopic effects). We note that the value of R_{int} of the X-ray diffraction data increased slightly from 0.0591 to 0.0598 after 10 times bending of the crystal which might indicate possible degradation. As shown in the panels a and b below, the crystal can clearly point to exactly the same position without being attracted by the magnet after 1000 cycles in the durability test. Taken together, the results from these experiments confirm that the bending of the crystal after many times may indeed have some effect on the quality of the crystal itself, however that effect is not detectable based on the reversibility of the deformation. Therefore, we have modified the language of the article.

The following text has been modified on page 9 line 188 of the revised manuscript:

Original text: “The crystal did not show any fatigue or defects, and pointed precisely at each point even after 3000 cycles (Figure 3f).”

Revised text: “The crystal of MNP//4 pointed precisely at each point under the control of the magnet even after 3000 cycles (Figure 3f).”

Response to the comments from Reviewer #2:

Overall Comment: *In the submitted article, the authors elegantly present an original attempt to integrate organic crystals into flexible actuators that can be remotely controlled using magnetic fields. Over the last decades, the field of the organic crystals popped-up with materials exhibiting interesting physical properties that paved the way towards potential brand-new actuators without clearly getting to the end of the fabrication and qualification of such devices. In the present paper, the authors successfully and elegantly bridge this gap and confirm the possibility to implement such crystals in conventional process flows that approach the standards of the traditional microfabrication techniques realm.*

As a preliminary observation I would suggest to the authors not to use the “soft robot” terminology; for an artefact to be qualified as a robot, it has to have the capacity to sense its environment, process the received information and act as a consequence of it. Instead of “robot”, I would use the “actuator” term, especially in the context of the paper.

I would split my report in three parts: experimental protocols, quantitative analysis and suggestions to improve the added value of the paper.

Response: The authors of this manuscript thank the Reviewer for the positive assessment and the useful suggestions. We agree with the reviewer’s argument that indeed “actuator” is a more appropriate term compared to “soft robot”, and this also makes the making the language more accurate and reasonable. Therefore, we have changed the phrase “soft robot” in the manuscript.

The following text has been modified on page 6 line 148 of the revised manuscript:

Original text: “Such a crystal can be considered equivalent to a soft robot, and was able to walk over a distance of up to 7 cm in 18 seconds.”

Revised text: “Such a crystal can be considered equivalent to a biomimetic actuator, and was able to move over a distance of up to 7 cm in 18 seconds”

Comment: 1.

Experimental protocols

All the chemistry and instrumentation protocols are thoroughly described either in the main article and in the supplementary information files. The fabrication of the actuators is clearly stated. It would be useful to provide the details of the magnetic particles spin coating protocol.

Response: We thank the Reviewer for this suggestion. Correspondingly, we have added all details on the chemistry and instrumentation to the Supporting Information, and we have also revised and added the details of the magnetic particles spin coating in the text. The following text has been added (page 2 lines 37) to the revised Supporting Information:

Added text: “Magnetic induction strength (B) was tested by (SG-3-A) gaussmeter. Magnet (diameter 36 mm, thickness 7 mm, vertical pull 6-29 kg).”

The following text has been modified on page 4 line 99 of the revised manuscript:

Original text: “This procedure was followed by spin-coating of one section of the polymer-coated crystals with iron(III) oxide magnetic nanoparticles (MNPs), whereby hybrid, partially MNP-coated crystals MNP//1–6 were obtained which had a small strip of MNP section closer to one end (Figure 1g).”

Revised text: “The polymer-coated crystals were then affixed at both ends to a capillary glass tube (Supplementary Figure 5), and a suspension of iron(III) oxide nanoparticles (MNPs) was aspirated with a syringe. One part of the polymer-coated crystals was spin-coated using the droplets at the tip of the syringe, whereby partially MNP-coated crystals of MNP//1–6 were obtained which had a small strip with MNP closer to one end (Figure 1g).”

New Supplementary Figure 5. Preparation of hybrid organic crystals. (a) PDDA/PSS_{8.5-4}. (b) MNP//4.

Comment: Six different organic compounds are used all over the experiments. Nevertheless, only some of them seem to be selected to perform some of the experiments (e.g. MNP//4 is used for the inchworm-like motility demonstration, MNP//4 again, as well as MNP//6, are used to somehow quantify the linear relationship between the magnetic force and the deformation of the hybrid structures). Could the authors comment on these selections? Is the MNP//4 structure more chemically responsive than the others?

Response: We thank the Reviewer for their comment and careful observation. The Reviewer has correctly noted that crystals used to showcase these motions were selected. This was done in order to make sure that they reflect the respective property. For example, MNP//4 was selected to demonstrate the inchworm-like motility, for 180 and 360-degree bending experiment, as well as for the fatigue resistance test because it is more slender and therefore more flexible compared to other crystals, and it could be bent along two different faces (see panels a and b in Supplementary Figure 4). Furthermore, MNP//6 was selected for the 180-degree bending tests, as well as to investigate the relationship between the magnitude of the magnetic force and the degree of bending test due to its observed pronounced flexibility. While, in principle, any of the crystals can be used for this purpose, in order to make sure that the thickness does not prevent the bending, we had to select slender, flexible specimens among the available crystals. Similarly, MNP//2, MNP//3 and MNP//4 were chosen for optical waveguide testing because they were observed to have better optical waveguide properties compared to some others samples. Altogether, for purpose of demonstration rather than optimization, we felt that manual selection of the crystals should be sufficient to be able to demonstrate these capabilities. As an example and to illustrate this strategy, in the revised version of the text we have added the advantages of 4 in the revised text (page 3 lines 92):

Added text: “A crystal of MNP//4, which was found to be capable of bending on two different crystal faces, was suspended into a round glass vessel, and a magnet was moved across the wall of the container.”

New Supplementary Figure 4. Photographs of crystal 4 bending in two different faces.

The following text has been modified on page 7 line 160 of the revised manuscript:

Original text: “A crystal of MNP//4 was placed suspended into a round glass vessel, and a magnet was moved along the wall of the container.”

Revised text: “A crystal of MNP//4, which was found to be capable of bending on two different crystal faces, was suspended into a round glass vessel, and a magnet was moved across the wall of the container.”

Comment: 2.

Quantitative analysis

An attempt to quantify the dependence of the hybrid structures deformation with respect to the magnetic field is given in the last part of the article and it clearly shows a linear dependence. What is less clear is the correlation, for instance, in between the positions shown in figure 5.b (eight of them) and the points plotted in figure 5.c (seven of them). Meanwhile, the extreme deformations would suggest a non-linear behavior of the tested structure while the plots still exhibit linear shapes. Could the authors be a bit more precise in their explanations regarding this part?

Response: We thank the Reviewer for pointing this out to us. The position on the far left from the eight positions in Figures 5b and 5d is the original state of MNP//4 unaffected by the magnet. The discussion focuses on the relationship between the magnetic field and the degree of bending of MNP//4, and therefore only seven points were plotted in Figure 5c and 5e (the leftmost position was not included). In order to make the discussion clearer, Figure 5b, 5d, and Supplementary Figures 13a and 13b were revised, and the position on the far left was removed. In addition, we discussed thoroughly and thought over to the possibility that the extreme deformations would indicate nonlinear behavior of the tested structure. Our quantitative linear fit of the detected magnetic induction intensity to the bending degree of the crystal by fixing the detection probe of a Gauss meter is indeed inadequate, and this experiment only shows that the bending degree of the crystal increases with the increase of the magnetic force within a certain range. Therefore, the relevant descriptions in the article and the related photographs have been modified.

Specifically, the following text has been modified on page 11 lines 246 of the revised manuscript:

Original text: “We were further interested in whether there is a quantitative relationship between the magnetic force and the degree of bending of the crystal, and to examine this we used a device shown in Figure 5a (Supplementary Figure 12). A hybrid crystal of MNP//4 was affixed at one end with glue, and it was slowly approached with a permanent magnet from one side perpendicular to its longest axis, while recording the magnetic induction intensity (B) with a probe connected to a Gaussmeter from the other side. In two experiments, the degree of bending of the crystal showed positive linear relationship with the magnetic force to some extent (Figure 5b–e). In order to test reproducibility, the position of the detection probe on the Gaussmeter was changed, and the measurements were repeated by using MNP//6 (Supplementary Figure 13). The results confirmed the correlation between the degree of bending of MNP//6 and the magnetic force.”

Revised text: “We were also interested in whether there is a relationship between the magnetic field strength and the degree of bending of the crystal. A setup shown in Figure 5a

(Supplementary Figure 18) was used to examine this hypothesis. A hybrid crystal of MNP//4 was affixed at one end with glue, and it was slowly approached with a permanent magnet from one side perpendicular to its longest axis, while recording the magnetic field strength (B) with a probe connected to a Gaussmeter from the other side. In two experiments, the bending of the crystal was found to increase with increasing magnetic force (Figure 5b–e). In order to test reproducibility, the position of the detection probe on the Gaussmeter was changed, and the measurements were repeated by using MNP//6 (Supplementary Figure 19). The results confirmed the relationship between the degree of bending of MNP//6 and the strength of the magnetic field. This observation provides further opportunities into modulation of the crystal deformation by using variable magnetic field.”

New Supplementary Figure 19. Relationship between the magnetic field intensity and crystal bending. (a,b) Images showing bending of MNP//6 crystals in test (a) and (b) while they were being approached with the magnet. (c,d) The ratio of the chord length (L) and arc length (C) of the crystal plotted against the magnetic field intensity (B).

New Figure 5. Experimental setup (a) and results (b–e) of effect of the magnetic field on crystal bending. (a) Schematic of the experimental setup. The magnet approaches a vertically standing crystal from the right side, and the deflection is recorded on the opposite side. (b,d) Images showing the bending of crystals of MNP//4 in the test (b) and (d) while they were being approached with the magnet. (c,e) The ratio of the chord length (L) and arc length (C) of the crystal plotted against the magnetic field strength (B).

Comment: *What would be interesting is to have an estimation of the effective Young modulus of the hybrid crystals by measuring the deformation and estimating the moment of inertia of the structures. This value is expected to be lower than the Young modulus of the sole crystals. I do not request these supplementary tests and data here as this is not the main scope of the paper. I am sure, given the high applicative potential of the presented structures, that a thorough qualification of actuation performances paper could come in the next future.*

Response: We agree with the Reviewer that estimating the effective Young's modulus of the hybrid crystal by measuring the deformation of the structure and estimating the moment of inertia of the structure is indeed very helpful to improve the high application potential of the presented structures. Our intention is to follow up on this work by an application-oriented study where the performance of the magnetically driven soft crystal actuators would be optimized in respect to both intrinsic and external factors, mainly the crystals' aspect ratio, length, external conditions, etc. These questions warrant a separate set of experiments and will be the focus of a follow-up work that will focus on thorough performance characterization.

Comment: 3.

Suggestions to improve the added value of the paper

Answering the questions asked above in the main body of the article will for sure improve its readability. My last comment will concern the optical waveguide part which is also well discussed and opportunely presented as one possible application. I would invite the authors to think bigger and eventually comment this in their concluding part: what if one the polymers (say, the external layer) is used as gas or biological sensor? The authors have already a device which has integrated actuation (magnetic) and sensing (optical) schemes which, by means of a well chose polymer, could be already used as biological or chemical ready for real world applications. This is, I believe, a huge added value of an already complete, well written, thoroughly documented and original paper which deserve publication in Nature Communications review.

Response: Thank you very much for your recognition of our work and for your valuable suggestions. As far as the current research status of flexible organic crystals is concerned, the use of other polymers to wrap flexible organic crystals for gas and biosensors has never been reported, except for the use of PVA polymers assembled on the outside of flexible organic crystals for humidity detection and sensors. The proposal of using polymers in combination with the properties of flexible organic crystals as gas and biosensors is indeed very promising and will contribute greatly to the future improvement of sensor versatility and sensitivity. We will try hard in our future work. The following text has been added to page 12 line 268 of the revised manuscript:

Added text: “Moreover, the strategy proposed to assemble polymers capable of different responses on the outer layer of flexible organic crystals can be a general approach to preparation of active sensing elements for detection of gases or biosensors, among other applications.”

.....

Response to the comments from Reviewer #3:

Overall Comment:

Dear Editor,

In this communication by “Anonymous” et al., the author/ authors present a novel and innovative way of bending slender and elastic molecular crystals using magnetic fields. As mentioned in the abstract and introduction, some molecular crystals exhibit elastic properties that allow them to be deformed via force, heat, or photons in the case of photomechanically responsive crystals. In this reviewer’s opinion, deforming crystals by light is still a superior method for shaping crystals since light can be focused, change wavelength, polarity, and intensity all in a flip of a switch, and can act on select regions of a crystal from unlimited distances without losing power, if properly focused. However, photomechanically responsive molecular crystals are scarce with larger photomechanical crystals that tend to fracture due to the buildup of interfacial stress between the photoproduct layer and reactant. In some cases, light cannot penetrate deep enough in the crystal due to the formation of an absorbing photoproduct layer. This paper presents a "general" method for coating cm-long acicular molecular crystals with alternating layers of poly(diallyldimethylammonium chloride) (PDDA) and a negatively charged polymer, poly(sodium 4-styrenesulfonate) (PSS). This composite layer acts as a platform onto which iron(III) oxide magnetic nanoparticles (MNPs) layer can be deposited. The method of coating these crystals with polymers followed by nanoparticles is subtle yet innovative. By coating, only one section or alternating sections of the molecular crystals studied they have created a truly hybrid new material that mechanically responds to a magnetic field. The mechanical response is large enough to be visualized with the added property of being fatigue-resistant over thousands of cycles at moderate temperatures. The mechanical response was assessed qualitatively and

quantitatively in air and water by varying the magnetic field intensity and measuring the deflection angle.

The abstract is concise and reflects the claims mentioned in the text and conclusion. The method used to measure the deflection angles was clearly stated in the experimental section. I highly recommend this for publication as communication in Nature Communications, however, there are some minor inconsistencies and suggestions that need to be addressed to improve the quality of this paper.

Please consider incorporating the following reference:

a) "Chiba H, Morimoto M, Irie M. Stepwise Assembly of Ultrathin Poly (vinyl alcohol) Films on Photoresponsive Diarylethene Crystals. Chemistry Letters. 2021 Jan 5;50(1):84-6." In this paper, Irie uses PVA to coat photoresponsive molecular crystals. However, his objective from coating these crystals was to prevent them from dissolving in organic solvents and not to detect water vapor. However, it is worth mentioning that people have thought of coating crystals with a polymer PVA before.

Response: We are very thankful to the Reviewer, who based on the thorough and astute comments apparently is a knowledgeable expert in this field, for recognizing the significance and the impact if the work presented in this manuscript, and for the generally positive assessment. The authors of this manuscript firmly believe that thorough and constructive comments are essential to improvement of the quality of the research work and its presentation, and in that regard, constructive comments are always welcome. Following the suggestion by the Reviewer, the reference that they pointed out was added in the revised version of the manuscript. Moreover, the following sentence was added in the introduction section (page 2 lines 45):

Added text: "Coating of organic crystals was used recently as a method to prevent their dissolution in organic solvents.⁴¹"

Comment: Other issues that need to be addressed:

1. What type of waveguides are these crystals? Are they the active or passive kind? Will the different types of waveguides (active or passive) respond differently in the presence of a polymer and MNP layer on the surface?

Response: We confirm that all crystals mentioned in the manuscript were tested as active optical waveguides that transmit their own fluorescence after excitation. In presence of polymer coating and MNPs on their surface, the crystals are able to transmit light both as active and passive waveguides. In order to clarify this point, we have revised the relevant sections of the manuscript. Specifically, the following text has been modified on page 9 line 209 of the revised manuscript:

Original text: "the optical loss coefficients (OLCs) of the original crystal 4 and MNP//4 were both tested and compared."

Revised text: "the optical loss coefficients (OLCs) of the original crystal 4 and MNP//4 were both tested and compared as active optical waveguides."

Moreover, the following text has been modified on page 10 line 219 of the revised manuscript:

Original text: "the optical transmission of the native and hybrid crystals 2 and 3 in their straight state were also recorded."

Revised text: "...the optical transmission of the native and hybrid crystals 2 and 3 in their straight state as active optical waveguides were also recorded (Supplementary Figure 15)."

The following text has been also added to page 10 line 222 of the revised manuscript:

Added text: "In addition, MNP//4 can be used as a passive optical waveguide for optical signal transmission (Supplementary Figure 17)."

New Supplementary Figure 17. Photographs of passive optical waveguide of crystal. (a) Crystal 4 in straight and bent states. (b) MNP//4 in straight and bent states.

Comment: 2. *Decorating select areas of acicular molecular crystals with MNP is applicable only when the crystals are long enough to handle with dexterous fingers. Can this process be automated or applied on much smaller crystals using specialized instruments such as AFM?*

Response: This is quite an impressive idea, and we thank the Reviewer for proposing it. In the current manuscript, we have been focusing on the coating of polymers and assembly of MNPs on needle-like organic crystals that are several centimeters long. The size of these crystals is therefore much larger than what would be practical to use AFM for coating of their tips, or any other automated techniques that are normally used for manipulation of smaller crystals. However, we are currently investigating the possibility for application of AFM as a tool to assemble active particles, and we hope to be able to make the results of these experiments public soon.

Comment: 3. *Are all the crystals studied isotropically elastic? It is worth highlighting that some crystals can be elastic along one crystal direct and not the other, therefore this method may or may not work on some crystals. This may be an added feature where if a crystal was bent in an undesirable direction it will snap.*

Response: This is an important point, and we thank the Reviewer for bringing it up. Indeed, some experiments require isotropically bending crystals. For example, in the 360-degree bending experiment in air, only the crystal MNP//4 meets the experimental requirements. The other crystals reported in this article cannot respond to this experiment because when in case they are bent to an undesirable direction, they will break. In order to clarify this important point, we have modified the manuscript and added a comment on this key point.

Specifically, the following text has been added (page 3 lines 92) to the revised manuscript:

Added text: “Form the studied materials, only crystals 4 could be bent on two of the crystals’ faces (Supplementary Figure 4).⁴³”

New Supplementary Figure 4. Photographs of crystal 4 bending in two different faces.

Moreover, the following text has been modified on page 7 line 160 of the revised manuscript:

Original text: “A crystal of MNP//4 was placed suspended into a round glass vessel, and a magnet was moved along the wall of the container.”

Revised text: “A crystal of MNP//4, which was found to be capable of bending on two different crystal faces, was suspended into a round glass vessel, and a magnet was moved across the wall of the container.”

Comment: 4. *Please elaborate on how you obtained these acicular crystals. Solvents used concentration, temperature, etc.*

Response: Following the suggestion, we have added the growth conditions of the acicular crystals to the text. The following text has been modified on page 3 line 88 of the revised manuscript:

Original text: “Long acicular crystals of all compounds were grown by solvent evaporation.”

Revised text: “Acicular crystals of all compounds were grown by solvent evaporation (Supplementary Figure 2).”

New Supplementary Figure 2. Photographs of crystals 1–6 at 365 nm. Fluorescence. Appropriate dichloromethane solutions of compounds 1, 2,3,4,5 and 6 were added to the test tubes respectively, where compound 1, 3,4 and 5 tubes were added with three times the volume of ethanol along the wall, compound 2 was added with two times the volume of petroleum ether along the wall and compound 6 was added with three times the volume of cyclohexane along the wall without destroying the surface of the previous solution. After one week or two weeks at room temperature, needle-like crystals 1–6 were obtained, respectively.

Comment: 5. *The fatigue resistance of the crystals was measured at ambient temperatures, was there any attempt to evaluate the resilience of the crystal and MNP layer at a slightly higher temperature?*

Response: Again, we thank the Reviewer for this suggestion. We have added a comment on the durability of hybrid organic crystals at slightly higher temperatures to the text. The following text has been added (page 9 lines 190) to the revised manuscript:

Added text: “The crystal of MNP//4 pointed precisely at each point under the control of the magnet even after 3000 cycles (Figure 3f). Moreover, the crystal showed high durability at slightly higher temperatures (50 °C) (Supplementary Figure 12). In addition, when the crystals of MNP//5 were placed in water for a different period of time, they were still able to bend by attraction by the magnet (Supplementary Figure 13).”

New Supplementary Figure 12. MNP//4 durability test at 50°C.

Comment: 6. *How come the polymer layer and MNPs did not fall off the crystal surface after actuation in water, knowing that the polymer layers are water-soluble? How long can the MNP-decorated crystal remain in the water?*

Response: This is also a very interesting and important aspect of the experimental of these hybrid materials. In order to verify the stability of the materials in these conditions, we placed the polymer-coated crystals in an aqueous solution at room temperature over different period of time, and we examined the surface by SEM. The results are shown in the figure below. It can be seen that when the polymer-coated crystals were left under in an aqueous solution for 7 days their surfaces did not show any deterioration. Moreover, we confirmed that the MNP- and polymer-modified crystals were still able to bend when approached by a magnet and after being placed in water for a long time. We have included the durability of the crystals placed in water for a long time in the revised version of the manuscript.

The following text has been added (page 9 lines 191) to the revised manuscript:

Added text: “In addition, when the crystals of MNP//5 were placed in water for a different period of time, they were still able to bend by attraction by the magnet (Supplementary Figure 13).”

New Supplementary Figure S13. Photograph of MNP//5 placed in water for different period of time.

Comment: *Minor inconsistencies:*

1. The geometrical isomer configurations of compounds 4, 5, and 6 depicted in Supplementary Scheme 3 are different from the ones in the main text. Please confirm the proper geometrical isomer from the crystal structure or $^1\text{H NMR}$.

Response: We thank the Reviewer for pointing out this minor yet important point. Accordingly, we have modified the geometrical isomer configurations of compounds 1, 4, 5 and 6 in Supplementary Scheme 3.

New Supplementary Scheme 1. The synthetic procedure of compound 1

New Supplementary Scheme 3. The synthetic procedures of compounds 2, 4, 5 and 6.

Comment: 2. *Supplementary Figure 2 was not helpful. This reviewer could not see any difference between the vials.*

Response: Thank the reviewer for pointing out this problem. In the revised version we have deleted the Supplementary Figure 2.

Comment: 3. *The ¹H NMR spectrum of compound 3 reveals the minor presence of a geometrical isomer. However, elemental analysis of compound 3 shows that it is very pure > 99.3%. Mention in the text that the minor presence of geometrical isomers in compound 3 will not affect the elemental analysis results since both isomers have the same molecular formula.*

Response: We find it quite necessary to mention in the text that the minor presence of geometrical isomers in compound 3 will not affect the results of the elemental analysis since both isomers have the same molecular formula. In the revised version of the manuscript, we have added this to the text. The following text has been added (page 3 lines 73) to the revised supplementary information:

Added text: “(the minor presence of geometrical isomers in compound **3** will not affect the elemental analysis results since both isomers have the same molecular formula).”

Comment: 4. *Figure 3a is not clear. Please point out with arrows where the crystal is located.*

Response: In the revised version, we have pointed out the position of the crystals in Figure 3a with arrows.

New Figure 3. Spatial control over the shape of the hybrid crystals by using magnetic field. (a) Optical photograph of an MNP//1–6 crystal bent at an arbitrary angle within a plane (side view). The blue arrow indicates the position of the crystal. (b) Schematic diagrams of MNP//1–6 bending at an arbitrary angle in a plane. (c) Photographs of a crystal of MNP//4 bending precisely towards 9 points that were marked on a silicon wafer. (d,e) Schematic diagrams of MNP//1–6 bending at an arbitrary angle in any direction in space (d), and in any direction in space inside a liquid (e). (f) Photographs of the MNP//4 crystal at different times in a durability test. During the test the crystal was bent between -90° and 90° by moving the magnet under the silicon wafer, followed by returning it to -90° . The blue arrow represents the direction of movement of the magnet.

Comment: *Conclusion:*

This paper is significant for both the crystal and polymer community since the authors were able to use both materials to produce a composite system that is greater than the sum of its parts. This method can be extended to coat photomechanical crystals to yield higher-order adaptronic crystals with more functions. I recommend publication after some minor revisions.

Response: We thank this Reviewer for extremely thorough assessment and manifold of very helpful comments that have helped in further improvement of the quality of our manuscript. Their time and patience with reading the manuscript and the supplementary materials are highly appreciated.

REVIEWERS' COMMENTS

Reviewer #1 (Remarks to the Author):

The authors have answered my concerns and questions in details. I recommend it to be published.

Reviewer #2 (Remarks to the Author):

From my point of view, all the issues have been thoroughly addressed by the authors. The manuscript is ready for publication.

Reviewer #3 (Remarks to the Author):

Dear Editor, all the necessary changes have been amended in the text. No objection to publish